# HEBBIAN GRAPH EMBEDDINGS

## ABSTRACT

Representation learning has recently been successfully used to create vector representations of entities in language learning, recommender systems and in similarity learning. Graph embeddings exploit the locality structure of a graph and generate embeddings for nodes which could be words in a language, products of a retail website; and the nodes are connected based on a context window. In this paper, we consider graph embeddings with an error-free associative learning update rule, which models the embedding vector of node as a non-convex Gaussian mixture of the embeddings of the nodes in its immediate vicinity with some constant variance that is reduced as iterations progress. It is very easy to parallelize our algorithm without any form of shared memory, which makes it possible to use it on very large graphs with a much higher dimensionality of the embeddings. We study the efficacy of proposed method on several benchmark data sets in Goyal & Ferrara (2018b) and favourably compare with state of the art methods. Further, proposed method is applied to generate relevant recommendations for a large retailer.

## 1 INTRODUCTION

Graph embeddings learn vector representations of nodes in a graph. [Cai et al. (2018)] and [Goyal & Ferrara (2018b)] give a comprehensive survey of graph embedding methods like node2vec [Grover & Leskovec (2016)] and also deep convolutional embeddings. The advantage of learning low dimensional embeddings is that they induce an order on the nodes of a graph which could be authors in a citation network, products in a recommender system, or words in an text corpus. The order could be established using an inner product or using another machine learning algorithm like a neural network or a random forest.

Our method uses error-free associative learning to learn the embeddings on graphs. The algorithm is quite simple, but very effective. We apply the learnt embeddings to the task of recommending items to users and to the task of link prediction and reconstruction.

Label propagation and message passing have been applied to many tasks like feature propagation [Heaukulani & Ghahramani (2013)], interest propagation, propagation of information in a population [Rapoport (1953)] and other network models of behavior like PageRank [Page et al. (1999)] and models of text like TextRank [Mihalcea & Tarau (2004)]. Instead of propagating a single unit of information, we propagate entire embeddings across the network. By propagating information on a graph iteratively, long distance similarities can also be learnt.

For link prediction and reconstruction, our results are directly comparable to [Goyal & Ferrara (2018b)]. We compare our results from the state of the art results in [Goyal & Ferrara (2018b)] in tables 1,5,6,7 and find that our result has significantly better results as compared to the techniques implemented in [Goyal & Ferrara (2018b)] and [Zhang & Chen (2018)] like SEAL, VGAE [Kipf & Welling (2016)], node2vec [Grover & Leskovec (2016)], GF (graph factorization) [Ahmed et al. (2013)], SDNE [Wang et al. (2016)], HOPE [Ou et al. (2016)], and LE [Belkin & Niyogi (2002)]. Our method is similar to LLE [Roweis & Saul (2000)] (we compare with algorithms that were developed after LLE) and our algorithm takes inspiration from simulated annealing which is theoretically sound and it iteratively reduces the global variance, and thus is able to outperform all of these algorithms. Note that LLE uses a loss function while our method is an instance of errorless learning which, as the results show, is more effective and embarrassingly parallelizable.

Annealing is a process that takes steel in a furnace from a very high temperature to gradu-

ally cooling to lower temperatures. This creates a self-organizing process that improves the ductility properties of steel. A high temperature implies higher variance. We take inspiration from this process in this paper in which initially, the variance is very high and is gradually reduced with the goal that the network gains more structure and finds a stable state after the iterations complete [Kirkpatrick et al. (1983)].

## 2 HEBBIAN GRAPH EMBEDDINGS

Hebbian learning is the simplest form of learning invented by Donald Hebb in 1949 in his book The organization of behavior [Hebb (1949)]. It is inspired by dynamics of biological systems. A synapse between two neurons is strengthened when the neurons on either side of the synapse (input and output) have highly correlated outputs. In essence, when an input neuron fires, if it frequently leads to the firing of the output neuron, the synapse is strengthened. In simple terms: "neurons that fire together wire together" [Hebb (1949)]. Recently, there's renewed interest in Hebbian learning. [Keysers & Gazzola (2014)] postulates that Hebbian learning predicts mirror-like neurons for sensations and emotions. [Treur (2016)] applies Hebbian learning in modelling of temporal-causal network.

Hebbian learning consists of a parameter update rule which is based on the strength of connection between two nodes, as applied to neural networks (based on firing tendencies of neurons on the opposite ends of a synapse). We extend the idea to graphs. Based on a pre-computed transition probability between two nodes, we update the parameters (the embeddings of a node) iteratively based on an error-free associative learning rule (nodes that are contextually connected should have similar embeddings, like word2vec for words [Mikolov et al. (2013)]). For a discussion on errorless learning, please see [McClelland (2006)].

We first initialize all embeddings to a multivariate normal distribution with mean 0 and variance $\sigma^2$.

$$w_j \sim N(0, \sigma^2 I) \tag{1}$$

We model the embedding at a node as a non-convex Gaussian mixture of the embeddings of the connected nodes. If there is an edge from node i to node j, the embedding of node j is modeled as follows:

$$w_j \sim N(w_i, \sigma^2 I) \tag{2}$$

The variance $\sigma^2$ starts off at a value of 10 and is divided by 1.1 every iteration in the spirit of simulated annealing [Kirkpatrick et al. (1983)]. The embedding of node j is updated as follows:

$$\tilde{\mathbf{w}}_i \sim N(w_i, \sigma^2 I) \tag{3}$$

$$\delta_j = \sum_i (\tilde{\mathbf{w}}_i * p_{ij} * \eta) \tag{4}$$

$$w_j = w_j + \delta_j \tag{5}$$

The $\delta_j$ are then simply added to the embedding at node j (where there is an edge from node i to node j). $p_{ij}$ is the transition probability and $\eta$ is the learning rate. The graph is weighted, asymmetric and undirected. Also, a random negative edge is selected at each node and the negative of the embeddings is propagated to both selected nodes with a fixed transition probability (we use 0.5). This iterative procedure learns the embeddings of all nodes in the graph and is able to generate very effective embeddings, as the next section shows. As shown in figure 1, the embeddings get propagated across the Gaussian graph iteratively.

$p_{ij} = (\#(i \rightarrow j))/(\#i)$

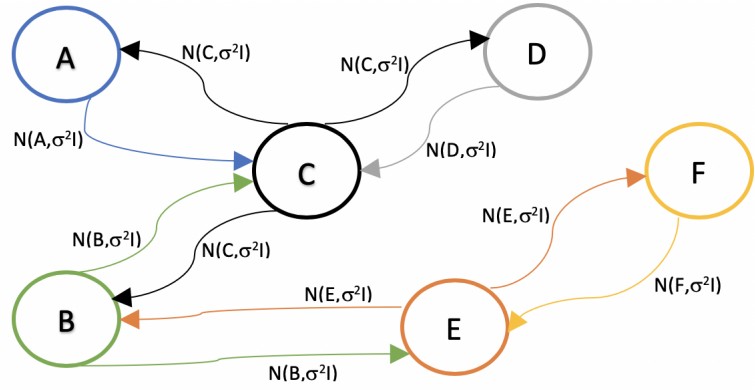

Figure 1: Propagation of Embeddings Across a Graph

---

**Algorithm 1** Hebbian Graph Embeddings

---

1: **procedure** FINDEMBEDDINGS($G$)
2:     Inputs: Weighted, asymmetric and undirected graph with nodes as nodes $(1, 2 \ldots, P)$ and edge weights as transition probabilities between nodes $p_{ij}$
3:     Hyper-parameters:
4:     $\sigma^2$ Variance of normal distribution (initial value = 10)
5:     $N$ Number of iterations of Hebbian learning
6:     $K$ Dimensionality of node representation
7:     $\tau$ Variance reduction factor (value = 1.1)
8:     Initialization: Initialize the nodes representation $w_i$ by sampling from a zero mean multivariate normal distribution $N(0, \sigma^2 I)$ of dimensionality $K$
9:     **for** each integer $m$ in $N$ **do**
10:       **for** each node $i$ in $P$ **do**
11:         **for** each node $j$ in $Adj(i)$ **do**

$$\tilde{\mathbf{w}}_i \sim N(\mathbf{w_i}, \sigma^{\mathbf{2}}\mathbf{I}) \tag{6}$$

$$\mathbf{w_j} \leftarrow \mathbf{w_j} + \eta\tilde{\mathbf{w}}_i p_{ij} \tag{7}$$

12:         **end for**
13:       **end for**

$$\sigma^{\mathbf{2}} \leftarrow \sigma^{\mathbf{2}}/\tau \tag{8}$$

14:     **end for**
15: **end procedure**

---

## 3   EXPERIMENTS AND RESULTS

We run our algorithm on three of the data sets used in [Goyal & Ferrara (2018b)] namely AstroPh, BlogCatalog and HepTh for both link prediction and reconstruction. Our algorithm outperforms several other algorithms that are implemented in [Goyal & Ferrara (2018b)] and [Goyal & Ferrara (2018a)]. We also compare our algorithm for link prediction using average precision and run-time with SEAL [Zhang & Chen (2018)] and VGAE [Kipf & Welling (2016)] in table 6 and table 7.

We start with an initial variance $\sigma^2$ of 10 and use the variance reduction factor $\tau$ of 1.1. We run the algorithm for 10 iterations. The algorithm is shown in *Algorithm 1*.

Link Prediction is the task of trying to predict a link between two nodes that were not part of the training data. Reconstruction tries to reconstruct the entire graph which is used entirely in the training set (i.e. there is no train/test separation).

Table 1: MAP Comparison with State of the Art for Reconstruction (see [Goyal & Ferrara (2018b)])

| Algorithm | Dimensionality | AstroPh | BlogCatalog | HepTh |
|---|---|---|---|---|
| Hebbian Graph Embeddings | 200 | 0.573 | 0.499 | 0.619 |
| node2vec | 256 | 0.56 | 0.24 | 0.42 |
| GF | 256 | 0.29 | 0.09 | 0.39 |
| SDNE | 256 | 0.46 | 0.33 | 0.5 |
| HOPE | 256 | 0.33 | 0.45 | 0.32 |
| LE | 256 | 0.26 | 0.09 | 0.4 |

Table 2: Mean Average Precision (MAP) results for network embeddings for Reconstruction of the entire graph

| DataSet | Nodes | Edges | Reconstruction Results of Varying Dimensionality | | | | | | | |
|---|---|---|---|---|---|---|---|---|---|---|
| | | | 10 | 20 | 50 | 100 | 200 | 300 | 400 | 500 |
| CondMat | 23,133 | 93,497 | 0.192 | 0.304 | 0.495 | 0.649 | 0.778 | 0.838 | 0.873 | 0.895 |
| GrQc | 5,242 | 14,496 | 0.245 | 0.407 | 0.625 | 0.763 | 0.860 | 0.894 | 0.910 | 0.918 |
| HepPh | 12,008 | 118,521 | 0.196 | 0.293 | 0.455 | 0.586 | 0.698 | 0.755 | 0.789 | 0.814 |
| AstroPh | 18,772 | 198,110 | 0.181 | 0.245 | 0.362 | 0.461 | 0.573 | 0.635 | 0.675 | 0.707 |
| HepTh | 27,770 | 352,807 | 0.188 | 0.261 | 0.402 | 0.509 | 0.619 | 0.679 | 0.709 | 0.732 |
| BlogCatalog | 10,312 | 333,983 | 0.432 | 0.432 | 0.458 | 0.491 | 0.499 | 0.507 | 0.508 | 0.496 |

We also run our algorithm on our recommender system and find that it is able to achieve a very high hit rate. Future work will focus more on the recommender system.

## 3.1 RESULTS ON RECONSTRUCTION

We ran our algorithm for reconstruction on publicly available data sets. Reconstruction tries to reconstruct the entire original graph (without splitting into train/test). As in [Goyal & Ferrara (2018b)], we sample 1024 nodes for calculation of the MAP. We run the algorithm for 10 iterations with a learning rate 1.0. The results in table 1, table 2 and figure 2 show that our algorithm is able to achieve good results on reconstruction when the dimensionality is large. As benchmarks, we use three data sets that [Goyal & Ferrara (2018b)] uses for reconstruction. Our results are favorably comparable on those three data sets. The other data sets are not used by [Goyal & Ferrara (2018b)] but the supporting code base as in [Goyal & Ferrara (2018a)] can be used to compare.

## 3.2 RESULTS ON THE RECOMMENDER SYSTEM OF A LARGE RETAILER

Also, in the recommender system at a large retailer, we used a sample of 200 thousand items as our population for training and measurement. 10% of the users are held out as the test set. The number of nodes in the graph is 200,000 and the number of edges is about 13.1 billion (note that the weight of an incoming edge might be different from an outgoing edge between any two nodes).

We measure the performance of our algorithm on the hit rate. Top 10 recommendations are generated per item based on the nearest neighbors of the generated embeddings based on an inner product (using all 200,000 items). Then, one random item from the users entire interaction history is chosen. Recommendations for this random item are computed. If any of the top 10 recommended items (other than the seed item) also occurs in the users interaction history, it is considered a hit. Otherwise a fail. The average hit rate is then the number of successes divided by the number of users in the test set. Results are shown in table 4. We use 10 iterations and a learning rate of 1.0.

The edges are determined using the induced graph from the consumer-product bipartite graph based on the co-viewing of the products. So, if two products were viewed by the same consumer, then we create an edge between them based on the same $p_{ij}$ weight between them (as described in section 2).

Table 3: Random Mean Average Precision (MAP) results (no training) for network embeddings for Reconstruction

| DataSet | Nodes | Edges | Random (no training) |
|---------|-------|-------|----------------------|
| | | | 500 (Dimension) |
| CondMat | 23,133 | 93,497 | 0.0139 |
| GrQc | 5,242 | 14,496 | 0.0126 |
| HepPh | 12,008 | 118,521 | 0.0233 |
| AstroPh | 18,772 | 198,110 | 0.0255 |
| HepTh | 27,770 | 352,807 | 0.0292 |
| BlogCatalog | 10,312 | 333,983 | 0.0364 |

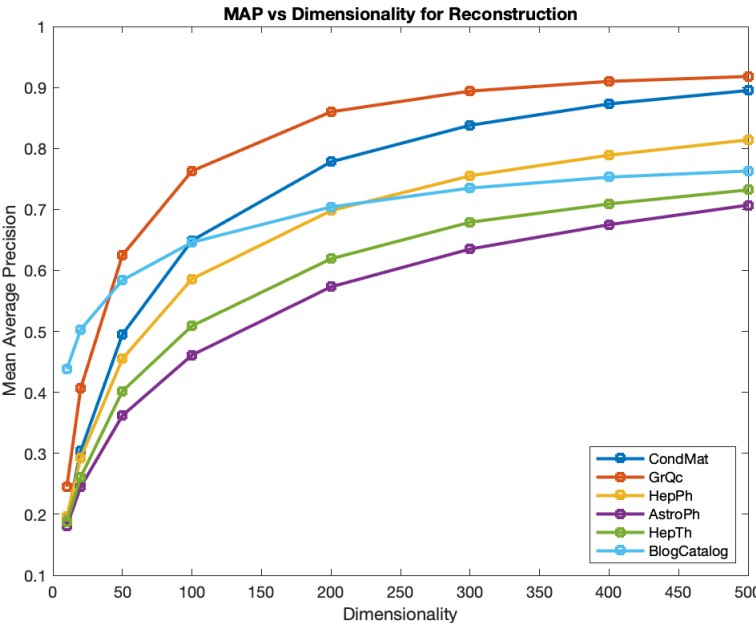

Figure 2: Mean Average Precision for Reconstruction with Varying Dimensionality.

## 3.3 RESULTS ON LINK PREDICTION

For link prediction, we use some of the data sets used in [Nickel & Kiela (2017)] and [Goyal & Ferrara (2018b)]. As in [Goyal & Ferrara (2018b)], we sample 1024 nodes for calculation of the MAP. We keep 10% of the edges as a held out test set. We run the algorithm for 10 iterations with a learning rate 1.0. The results in table 8, table 9 and figure 3 show that our algorithm is able to achieve good results on link prediction when the dimensionality is large. As benchmarks, we use three data sets that [Goyal & Ferrara (2018b)] uses for link prediction. Our results are favorably comparable on those three data sets for link prediction. [Goyal & Ferrara (2018b)] also has a supporting code base [Goyal & Ferrara (2018a)] which can be used to compare on other data sets.

We also compare our algorithm with SEAL and VGAE and find that our algorithm outperforms VGAE on all four data-sets and outperforms SEAL on one of the four data-sets. Note that since our algorithm is an Apache Spark application, there is some initial time spent on initialization and allocation of resources. The larger the graph, the more noticeable is the difference in the run-time. For instance, it might be infeasible to run SEAL or VGAE on our recommender system data-set with 200,000 nodes and 13.1 billion edges.

It is quite easy to parallelize the algorithm, and we implement it on Apache Spark. We run the algorithm for 10 iterations (which takes about 3 hours on the parallel implementation on recommender

Table 4: Results on a very large graph for recommender systems at a large retailer

| Dimensionality | HitRate@10 |
|---|---|
| 100 | 24.2% |
| 200 | 30.1% |
| 250 | 31.1% |

Table 5: MAP Comparison with State of the Art for Link Prediction (see [Goyal & Ferrara (2018b)])

| Algorithm | Dimensionality | AstroPh | BlogCatalog | HepTh |
|---|---|---|---|---|
| Hebbian Graph Embeddings | 200 | 0.317 | 0.202 | 0.339 |
| node2vec | 256 | 0.025 | 0.17 | 0.04 |
| GF | 256 | 0.15 | 0.02 | 0.17 |
| SDNE | 256 | 0.24 | 0.19 | 0.16 |
| HOPE | 256 | 0.25 | 0.07 | 0.17 |
| LE | 256 | 0.21 | 0.04 | 0.23 |

system data and from 5 minutes to 2 hours (depending on the dimensionality) on the publicly available data). We found that the learning rate does not affect the results in any significant way (we use 1.0).

## 4 CONCLUSION

In this paper, we described a simple, but very effective algorithm to learn the embeddings on a graph. The results show that the algorithm, as applied to the tasks of link prediction and reconstruction, is able to perform well when the dimensionality of the embeddings is large. This shows the effectiveness of learning on graphs using iterative methods. Its a useful experiment of error-free (errorless) learning on graphs. Our method can learn long distance similarities because of the iterative nature of the algorithm which percolates the embeddings on the weighted graph.

A distinctive advantage of our approach is that it is very easy to parallelize the algorithm without any need for shared memory. It is quite easy to implement the algorithm on platforms like Apache Spark, which makes the algorithm amenable to very large graphs which cannot be processed on one machine.

Our recommender system work was tested live and it did very well. But because our item graph has a very large number of nodes and edges, we omit the implementation of [Nickel & Kiela (2017)] and [Goyal & Ferrara (2018b)] for our recommender system.

Other algorithms like in [Vinh et al. (2018)] and [Chamberlain et al. (2019)] could be compared with our work. There is still an opportunity to improve the algorithm through hyperparameter tuning. It might be interesting to measure the algorithm with a much higher dimensionality of the embeddings.

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

Table 6: Comparison of Average Precision with other State of the Art algorithms for Link Prediction (see [Zhang & Chen (2018)] for more details), randomly chosen 10% of edges held out as the test set

| Algorithm | Power | PB | USAir | C.Ele |
|---|---|---|---|---|
| Hebbian Graph Embeddings | 93.11 | 93 | 95.92 | 84.99 |
| SEAL | 86.69 | 94.55 | 97.13 | 88.81 |
| VGAE | 75.91 | 90.38 | 89.27 | 78.32 |

Table 7: Comparison of Run-Time (seconds) for Link Prediction (see [Zhang & Chen (2018)] for more details)

| Algorithm | Power | PB | USAir | C.Ele |
|---|---|---|---|---|
| Hebbian Graph Embeddings | 287 | 233 | 237 | 172 |
| SEAL | 1640 | 146 | 31 | 16 |

Benjamin Paul Chamberlain, Stephen R Hardwick, David R Wardrope, Fabon Dzogang, Fabio Daolio, and Saúl Vargas. Scalable hyperbolic recommender systems. *arXiv preprint arXiv:1902.08648*, 2019.

Palash Goyal and Emilio Ferrara. Gem: A python package for graph embedding methods. *J. Open Source Software*, 3(29):876, 2018a.

Palash Goyal and Emilio Ferrara. Graph embedding techniques, applications, and performance: A survey. *Knowledge-Based Systems*, 151:78–94, 2018b.

Aditya Grover and Jure Leskovec. node2vec: Scalable feature learning for networks. In *Proceedings of the 22nd ACM SIGKDD international conference on Knowledge discovery and data mining*, pp. 855–864. ACM, 2016.

Creighton Heaukulani and Zoubin Ghahramani. Dynamic probabilistic models for latent feature propagation in social networks. In *International Conference on Machine Learning*, pp. 275–283, 2013.

Donald Olding Hebb. *The Organization of Behavior*. Wiley & Sons, 1949.

Christian Keysers and Valeria Gazzola. Hebbian learning and predictive mirror neurons for actions, sensations and emotions. *Philosophical Transactions of the Royal Society B: Biological Sciences*, 369(1644):20130175, 2014.

Thomas N Kipf and Max Welling. Variational graph auto-encoders. *arXiv preprint arXiv:1611.07308*, 2016.

Scott Kirkpatrick, C Daniel Gelatt, and Mario P Vecchi. Optimization by simulated annealing. *science*, 220(4598):671–680, 1983.

James L McClelland. How far can you go with hebbian learning, and when does it lead you astray? *Processes of change in brain and cognitive development: Attention and performance xxi*, 21: 33–69, 2006.

Rada Mihalcea and Paul Tarau. Textrank: Bringing order into text. In *Proceedings of the 2004 conference on empirical methods in natural language processing*, pp. 404–411, 2004.

Tomas Mikolov, Kai Chen, Greg Corrado, and Jeffrey Dean. Efficient estimation of word representations in vector space. *arXiv preprint arXiv:1301.3781*, 2013.

Maximillian Nickel and Douwe Kiela. Poincaré embeddings for learning hierarchical representations. In *Advances in neural information processing systems*, pp. 6338–6347, 2017.

Table 8: Mean Average Precision (MAP) results for network embeddings for Link Prediction (10% randomly chosen edges are held out as the test set)

| DataSet | Nodes | Edges | Link Prediction Results for Varying Dimensionality | | | | | | | |
|---|---|---|---|---|---|---|---|---|---|---|
| | | | 10 | 20 | 50 | 100 | 200 | 300 | 400 | 500 |
| CondMat | 23,133 | 93,497 | 0.070 | 0.130 | 0.251 | 0.350 | 0.450 | 0.507 | 0.531 | 0.544 |
| GrQc | 5,242 | 14,496 | 0.064 | 0.129 | 0.233 | 0.292 | 0.332 | 0.348 | 0.363 | 0.383 |
| HepPh | 12,008 | 118,521 | 0.065 | 0.121 | 0.213 | 0.289 | 0.346 | 0.384 | 0.401 | 0.424 |
| AstroPh | 18,772 | 198,110 | 0.060 | 0.092 | 0.179 | 0.235 | 0.317 | 0.357 | 0.388 | 0.409 |
| HepTh | 27,770 | 352,807 | 0.070 | 0.120 | 0.203 | 0.259 | 0.339 | 0.370 | 0.383 | 0.407 |
| BlogCatalog | 10,312 | 333,983 | 0.183 | 0.182 | 0.198 | 0.198 | 0.202 | 0.217 | 0.210 | 0.212 |

Table 9: Random Mean Average Precision (MAP) results (no training) for network embeddings for Link Prediction

| DataSet | Nodes | Edges | Random (no training) |
|---|---|---|---|
| | | | 500 (Dimension) |
| CondMat | 23,133 | 93,497 | 0.007 |
| GrQc | 5,242 | 14,496 | 0.007 |
| HepPh | 12,008 | 118,521 | 0.010 |
| AstroPh | 18,772 | 198,110 | 0.009 |
| HepTh | 27,770 | 352,807 | 0.009 |
| BlogCatalog | 10,312 | 333,983 | 0.014 |

Mingdong Ou, Peng Cui, Jian Pei, Ziwei Zhang, and Wenwu Zhu. Asymmetric transitivity preserving graph embedding. In *Proceedings of the 22nd ACM SIGKDD international conference on Knowledge discovery and data mining*, pp. 1105–1114. ACM, 2016.

Lawrence Page, Sergey Brin, Rajeev Motwani, and Terry Winograd. The pagerank citation ranking: Bringing order to the web. Technical report, Stanford InfoLab, 1999.

Anatol Rapoport. Spread of information through a population with socio-structural bias: I. assumption of transitivity. *The bulletin of mathematical biophysics*, 15(4):523–533, 1953.

Sam T Roweis and Lawrence K Saul. Nonlinear dimensionality reduction by locally linear embedding. *science*, 290(5500):2323–2326, 2000.

Jan Treur. *Network-oriented modeling*. Springer, 2016.

Tran Dang Quang Vinh, Yi Tay, Shuai Zhang, Gao Cong, and Xiao-Li Li. Hyperbolic recommender systems. *arXiv preprint arXiv:1809.01703*, 2018.

Daixin Wang, Peng Cui, and Wenwu Zhu. Structural deep network embedding. In *Proceedings of the 22nd ACM SIGKDD international conference on Knowledge discovery and data mining*, pp. 1225–1234. ACM, 2016.

Muhan Zhang and Yixin Chen. Link prediction based on graph neural networks. In *Advances in Neural Information Processing Systems*, pp. 5165–5175, 2018.

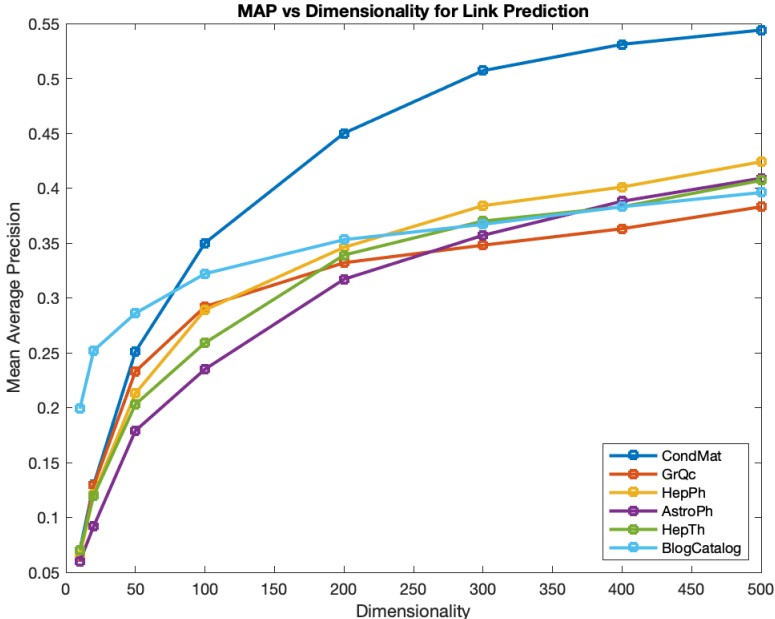

Figure 3: Mean Average Precision for Link Prediction with Varying Dimensionality.

