# OpenReview forum: "Hebbian Graph Embeddings"
_ICLR.cc/2020/Conference — Reject_

### Official Review · AnonReviewer1 · 2019-10-22
**Official Blind Review #1**

**Rating:** 1

**Review:**

In this paper, the authors proposed a simple but effective node embedding method for large-scale graphs.
The proposed method is based on Hebbian learning, enhancing the connections between neighbor nodes iteratively.
The idea is very straightforward and suitable for large-scale applications.
The authors tested the proposed method on multiple real-world datasets under different configurations.

The strategy of the work makes sense to me, but the work itself is incomplete.
As mentioned in the conclusion, there are some potential competitors of the proposed method, which should be considered as baselines in the experiments.
Additionally, for conceptual proof, the authors can consider a synthetic/real-world dataset with relatively small size and compare their method with state-of-the-art methods.

Additionally, the notations in Eqs. (1-3) are inconsistent with those in Algorithm 1. Especially Eq.(3), delta_j should be an incremental vector of node j’s embedding, while N(I, sigma^2 I) is a distribution. The authors should write Eqs. (1-3) as Eqs. (4-6).

**Experience Assessment:**

I have published one or two papers in this area.

**Review Assessment: Checking Correctness Of Derivations And Theory:**

I carefully checked the derivations and theory.

**Review Assessment: Checking Correctness Of Experiments:**

I carefully checked the experiments.

**Review Assessment: Thoroughness In Paper Reading:**

I read the paper thoroughly.

---

> ### Author Response · Authors · 2019-11-06
> **Revised paper**
>
> Thank you for the comments!
> We have added results from other state of the art algorithms like mentioned in [1].
> We find that our algorithm outperforms others.
> Please see table 1 and table 5 in the modified paper.
> The notation has been corrected.
>
> [1] Graph embedding techniques, applications, and performance: A survey

---

### Official Review · AnonReviewer3 · 2019-10-24
**Official Blind Review #3**

**Rating:** 1

**Review:**

[Summary]
This paper proposes an error-free rule update method for graph embedding. The authors employ the Hebbian learning concept iteratively using the pre-calculated transition probability. They evaluate their model on six benchmark datasets.

[Pros]
- Very simple and fast by using an error-free update rule.

[Cons]
- The introduction is not organized. What are the motivation, related work, and contribution?
- No comparison with conventional methods such as PageRank and NN-based models such as SEAL [1] and VGAE [2]. Even if this method is not learning-based, the proposed model should be compared.
- The authors claimed they applied their model to real recommendation systems. But there is no specific information. At least, the author describes what is recommended, the data size, how large performance is improved, etc.
- It is required to be evaluated on conventional datasets.
- Ablation on sigma

[1] M. Zhang and Y Chen, Link Prediction Based on Graph Neural Networks, NIPS 2018.
[2] Thomas N Kipf and Max Welling. Variational graph auto-encoders. arXiv preprint arXiv:1611.07308, 2016.



**Experience Assessment:**

I have published one or two papers in this area.

**Review Assessment: Checking Correctness Of Derivations And Theory:**

I assessed the sensibility of the derivations and theory.

**Review Assessment: Checking Correctness Of Experiments:**

I assessed the sensibility of the experiments.

**Review Assessment: Thoroughness In Paper Reading:**

I read the paper at least twice and used my best judgement in assessing the paper.

---

> ### Author Response · Authors · 2019-11-06
> **Revised paper**
>
> Thank you for the comments!
> We have updated the introduction.
> We have added results from other state of the art algorithms like mentioned in [1].
> We find that our algorithm is significantly better.
> Please see table 1 and table 5 in the modified paper.
>
> [1] Graph embedding techniques, applications, and performance: A survey

---

> > ### Comment · AnonReviewer3 · 2019-11-10
> > **Thank you for your response.**
> >
> > I appreciate the authors' efforts.
> >
> > However, [1] was published in May 2017. For 2.5 years, many advanced models for graph representation learning have been proposed. Therefore, I recommended comparing the proposed method with recent works such as [2] for the link prediction. Also, VGAE [3] was introduced as related work in [1], but [1] did not provide the comparison results with [3]. Considering that the main task of this work is graph reconstruction, at least, the proposed method should be compared with these methods and their competitiveness.
> >
> > Also, it seems that the proposed method has an advantage in its computation time. So, it is important to compare real computation times for training and inference with those of other models.
> >
> > Considering the characteristics of ICLR (learning representation), if the proposed method makes a good representation, the authors need to investigate how the method can do that with intensive analysis including representation visualization.
> >
> > [1] Goyal and Ferrara, Graph embedding techniques, applications, and performance: A survey, ArXiv 2017.
> > [2] M. Zhang and Y Chen, Link Prediction Based on Graph Neural Networks, NIPS 2018.
> > [3] Thomas N Kipf and Max Welling. Variational graph auto-encoders. arXiv preprint arXiv:1611.07308, 2016.

---

> > > ### Author Response · Authors · 2019-11-10
> > > **Comparison with SEAL and VGAE**
> > >
> > > Thank you for the response!
> > >
> > > We ran our algorithm for comparison with SEAL and VGAE.
> > > Our algorithm outperforms VGAE on all four data-sets.
> > > And, our algorithm outperforms SEAL on one out of four data sets.
> > > Please see tables 6 and 7.
> > > Also, note that our algorithm is run on Apache Spark, so there is some initial time spent on initialization and resource allocation.
> > > The larger the graph, the more noticeable is the difference in the run-time. Please see the power data-set.
> > > It might also be infeasible to run SEAL and VGAE on our recommender system graph with 200,000 nodes and 13.1 billion edges.
> > > Please review the modified paper. Thanks again.

---

### Official Review · AnonReviewer2 · 2019-10-24
**Official Blind Review #2**

**Rating:** 1

**Review:**

Thanks to the authors for their response. There are still significant issues with motivation, writing, and baseline comparisons (the latter noted by R3). I would encourage the authors to continue to polish and investigate their method and submit to a future conference.

=====

This paper proposes an approach to learning embeddings associated with nodes in a graph. Inspired by Hebbian learning, the representations of a node are iteratively updated to be similar to representations of its neighbors. The update is performed by adding a scaled vector sampled from a Gaussian centered on a neighbor's representation to the current node's vector. After learning, the embeddings may be used for tasks such as graph reconstruction, link prediction, or product recommendation.

Contributions include:
* Proposal of an approach to learn embeddings of nodes in a graph in which representations of a node are updated by scaled and Gaussian-perturbed versions of its neighbors' representations.
* Experiments demonstrating reconstruction and link prediction performance of the proposed approach on several datasets, as well as product recommendation perfomance on a large retail dataset.

There are several significant concerns with the paper as it currently stands. The three most pressing issues are as follows:
1. The proposed algorithm is not situated relative to related work.
2. The paper does not provide the reader with enough details or precision to be able to replicate the work.
3. Experiments do not compare to any baselines other than random embeddings.

The paper suggests that the proposed algorithm is a form of Hebbian learning because the representation of nearby nodes in the graph are encouraged to be similar. However, this idea has long been used for learning node embeddings (for example, LLE encourages representations of a node to be predicted as a linear combination of neighboring nodes). The connection seems loose other than a superficial similarity in the update rule and naming the algorithm after Hebbian learning is somewhat misleading.

The algorithm is reminiscent of message-passing inference in a continuous Markov random field with pairwise potentials encouraging nearby nodes to have similar representations. I am not an expert in graph embedding approaches, but I would be surprised if the approach could not be easily related to classical approaches such as MDS or LLE.

There are several notational/clarity issues:
* j is used for both the node index and the embedding of the node itself (equation 1-2). Replacing the j on the left hand side with w_j would resolve ambiguity and bring the equations in line with Algorithm 1.
* Equation 2 is inconsistent with equation 1 because they both specify different distributions over the same embedding. Framing equation 1 as an initialization and equation 2 as providing the conditional distribution p(w_j | w_i) may make the situation clearer.
* Equation 3 is a mixture of Gaussian distributions, yet \delta_j is a vector added to the current node embedding. Instead, first write \tilde{w}_i as a sample from the Gaussian and then let \delta_j be the weighted sum of the samples.
* Equation 3 is not consistent with equations 4-5. Equation 3 suggests that a node is updated by summing over neighbors and then applying the update. But Algorithm 1 suggests that nodes are updated based on only a single neighbor at a time.
* What does it mean when the negative embedding is propagated with a small transition probability? This should be described mathematically.
* It is misleading to call the graph a Gaussian hierarchy, since a hierarchy implies that certain nodes are higher than others.
* How are the "transition probabilities" set for an unweighted graph? Specifically, the GrQc dataset doesn't appear to have edge weights.
* What values of the variance and \tau hyperparameters were used?
* How are reconstructions and link predictions computed?

Experimentally, the proposed approach is not compared to any baselines other than random embeddings. The claim made in the paper that the method compares favorably is thus not backed up by results. The results in section 3.2 should be described in greater detail. If the items are nodes, then how are edges and weights determined?

 Other specific comments:
* What is the connection between the current work and hyperbolic geometry of Nickel & Kiela (2017)? The proposed algorithm does not rely on hyperbolic geometry so this seems like a non sequitur.
* Algorithm 1: Rather than describing the algorithm in terms of the intended application (products), it would be useful to describe it in general terms and then use retail products as specific application.
* Figures 2 and 3 are not particularly useful. The most important information for the reader or practitioner is how various methods compare on the same dataset, not how a single method performs across different datasets.

Questions for the authors:
* How is the proposed algorithm similar/different to related approaches for learning node embeddings?
* What are baseline results for related algorithms on the datasets experimented upon?
* What is the role of the variance scaling? How do the results change if the variance is reduced to 0 immediately after random initialization?

**Experience Assessment:**

I do not know much about this area.

**Review Assessment: Checking Correctness Of Derivations And Theory:**

I assessed the sensibility of the derivations and theory.

**Review Assessment: Checking Correctness Of Experiments:**

I assessed the sensibility of the experiments.

**Review Assessment: Thoroughness In Paper Reading:**

I read the paper at least twice and used my best judgement in assessing the paper.

---

> ### Author Response · Authors · 2019-11-06
> **Revised paper with comparison to state of the art**
>
> Thank you for the comments!
> We have added results from other state of the art algorithms like mentioned in [1].
> We find that our algorithm outperforms all others in [1].
> Please see table 1 and table 5 in the modified paper.
> The key contribution of our algorithm is the inspiration from annealing and errorless learning.
> We were not aware of LLE. We have added a reference and some notes on how our paper is different.
> Note that LLE uses a loss function while our method is error-free or Hebbian; this is an important difference.
> For this reason, our method is embarrassingly parallelizable on platforms like Apache Spark.
> All other comments have been noted and we have made changes to the paper respectively.
>
> [1] Graph embedding techniques, applications, and performance: A survey

---

### Decision · Program_Chairs · 2019-12-19

**Decision:**

Reject

**Comment:**

The paper learns an embedding on the nodes of the graph, iteratively aligning the vector associated to a node with that of its neighbor nodes (based on the Hebbian rule).

The reviews state that the approach is interesting though very natural/straightforward, and that it might go too far to call it "Hebbian" (Rev#2) - you might want also to see it as a Self-Organizing Map for graphs.

A main criticism was about the comparison with the state of the art (all reviewers). The authors did add empirical comparisons with the suggested VGAE and SEAL, and phrase it nicely as "our algorithm outperforms SEAL on one out of four data sets". Looking at the revised paper, this is true: the approach is outperformed by SEAL on 3 out of 4 datasets.

Another criticism regards the insufficient analysis of the results (e.g. through visualization, studying the clusters obtained along different runs, etc).
This aspect is not addressed in the revised version.

An excellent point is the scalability of the approach, which is worth emphasizing.

I thus encourage the authors to rewrite and polish the paper, improving the positioning of the proposed approach w.r.t. the state of the art, and providing a more thorough analysis of the results.